# Snow Cover in the Three Stable Snow Cover Areas of China and Spatio-Temporal Patterns of the Future

**Yifan Zou** [1], **Peng Sun** [1,*] , **Zice Ma** [1], **Yinfeng Lv** [1] and **Qiang Zhang** [2]

1   School of Geography and Tourism, Anhui Normal University, Wuhu 241002, China; 431379203@ahnu.edu.cn (Y.Z.); mzc19950320@ahnu.edu.cn (Z.M.); 939044955@ahnu.edu.cn (Y.L.)
2   Academy of Disaster Reduction and Emergency Management, Faculty of Geographical Science, Beijing Normal University, Beijing 100875, China; zhangq68@bnu.edu.cn
*   Correspondence: sun68peng@ahnu.edu.cn

**Abstract:** In the context of global warming, relevant studies have shown that China will experience the largest temperature rise in the Qinghai–Tibet Plateau and northwestern regions in the future. Based on MOD10A2 and MYD10A2 snow products and snow depth data, this study analyzes the temporal and spatial evolution characteristics of the snow cover fraction, snow depth, and snow cover days in the three stable snow cover areas in China, and combines 15 modes in CMIP6 snow cover data in four different scenarios with three kinds of variables, predicting the spatiotemporal evolution pattern of snow cover in China's three stable snow cover areas in the future. The results show that (1) the mean snow cover fraction, snow depth, and snow cover days in the snow cover area of Northern Xinjiang are all the highest. Seasonal changes in the snow cover areas of the Qinghai–Tibet Plateau are the most stable. The snow cover fraction, snow depth, and snow cover days of the three stable snow cover areas are consistent in spatial distribution. The high values are mainly distributed in the southeast and west of the Qinghai–Tibet Plateau, the south and northeast of Northern Xinjiang, and the north of the snow cover area of Northeast China. (2) The future snow changes in the three stable snow cover areas will continue to decline with the increase in development imbalance. Snow cover fraction and snow depth decrease most significantly in the Qinghai–Tibet Plateau and the snow cover days in Northern Xinjiang decrease most significantly under the SSPs585 scenario. In the future, the southeast of the Qinghai–Tibet Plateau, the northwest of Northern Xinjiang, and the north of Northeast China will be the center of snow cover reduction. (3) Under the four different scenarios, the snow cover changes in the Qinghai–Tibet Plateau and Northern Xinjiang are the most significant. Under the SSPs126 and SSPs245 scenarios, the Qinghai–Tibet Plateau snow cover has the most significant change in response. Under the SSPs370 and SSPs585 scenarios, the snow cover in Northern Xinjiang has the most significant change.

**Keywords:** MODIS; CMIP6; snow cover fraction; snow depth; snow cover days; Qinghai–Tibet Plateau; Northern Xinjiang; Northeast China

## 1. Introduction

Snow is an important part of the cryosphere. Because of its uniquely strong reflectivity, weak thermal conductivity, and heat absorption during melting, snow has a huge impact on energy balance, atmospheric circulation, and the hydrological cycle [1,2]. Snow is the most sensitive environmental change response factor in the cryosphere and an indicator of global climate change [3]. The International Panel on Climate Change (IPCC) AR6 assessment report has a more accurate estimate of global warming [4]. The shrinking rate of the cryosphere is increasing. According to the IPCC AR5 assessment, the snow cover range and snow cover in the northern hemisphere are decreasing more obviously, and the snow cover reduction in spring is the most prominent [5]. Snow will cause a series of natural disasters in the process of melting [6]. Therefore, in order to make rational use of snow

resources, prevent natural disasters, and ensure agricultural production, it is necessary to study the temporal and spatial changes of snow [7].

The snow cover fraction (SCF), snow depth (SD), and snow cover days (SCDs) are some of the important indicators in snow cover studies. The SD and SCDs are also indicators of the climate and environmental characteristics of snow cover and water resources conditions [8–10]. On the global and continental scales, snow cover in high-altitude mountains and in spring and summer showed a clear downward trend [11–13]. China is rich in snow resources and is also a popular area for snow studies. The study of snow in China is mainly distributed across three stable snow cover areas [14,15]. Under the context of global warming, 60% of the regions in the Qinghai–Tibet Plateau (TP) have a downward trend in SCDs, and the decline of the SCF is most obvious in summer [16,17]. With the shrinking of glaciers and the increase in snow meltwater, high-altitude areas with fragile ecological environments, especially the Hengduan Mountains, will have structural instability, which will lead to a series of natural disasters, such as landslides, mudslides, and floods [18]. Disasters such as ice avalanches and glacial lake outbursts will also occur in areas above 4000 m [19]. The SCDs of Northern Xinjiang (NX) have shown an insignificant decreasing trend, the SCF have shown a significant seasonal change below 4000 m, and the SD of the Tianshan Mountains has shown a significant decreasing trend [9,15,20,21]. The summer snow meltwater in NX is also an important resource for agricultural production in this area. The melting of snow provides replenishment for the river and important domestic water for arid areas and has an impact on agricultural production activities [22–24]. The SD in Northeast China (NC) is increasing, and the SCDs are decreasing [16,25]. The melting of snow in spring in NC leads to changes in soil water and affects the summer climate [26].

Snow is mostly distributed in sparsely populated high-altitude mountainous areas and cold high-latitude regions, especially in the three stable snow cover areas of China. Restricted by terrain and climatic conditions, the construction of meteorological stations is difficult, resulting in fewer meteorological stations in snow-rich areas, especially west of the TP [27]. It is difficult to carry out snow-related studies on a large scale. The spatial resolution of passive microwave remote sensing data is low, and it is impossible to accurately observe the regional snow [8,28]. However, MODIS data have the advantages of easy access, low cost, continuous observation time, and a large monitoring area [29,30]. MODIS also has a good monitoring effect in mountainous areas [31,32]. In previous studies, it has been confirmed that the eight-day snow product MOD10A2 has a better effect on snow observation than the daily snow product MOD10A1 [33,34]. Therefore, it was more accurate and feasible to use MOD10A2 to study snow cover in this study.

Although remote sensing can easily and quickly obtain snow cover information, due to the limitation of the time series length, rough spatial resolution, and cloud layer interference, it is necessary to use models to invert snow cover. At present, there are many studies on snow models. Wobus et al. [35] used the Utah energy balance model to simulate the winter ski areas in the United States. The advantages of the UEB model are high computational efficiency, few input parameters, and reliable results. The research shows that the UEB model has high accuracy in the simulation of snow depth in ski areas. Collados et al. [36] used an improved cellular automata (CA) model to study the snow cover area of the Sierra Nevada mountains. The CA model calculated the snow cover area through five parameters and two driving variables and corrected each parameter to calculate the snow cover area. Adjusted to optimal conditions, the results of the study suggest a significant reduction in snow cover in the region in the future. The Coupled Model Intercomparison Project (CMIP) initiated by the World Climate Research Program (WCRP) is a new generation of experimental data. CMIP6 can better reveal the changes caused by natural non-forcing, and the response to changes in radiative forcing in a multi-mode background [37]. Compared to the previous generation, CMIP5, it is more accurate in snow recognition, especially in winter snow monitoring [38,39]. Currently, there are few studies on snow cover prediction in China's three stable snow cover areas using CMIP6 model data. CMIP5 has mainly been used to study future snow cover changes. The CMIP5 snow cover change studies have

found that the SCF of the northern hemisphere land under four different scenarios will be reduced by 7.2–24.7% in the future, which is closely related to climate change. The spring snow cover in the northern hemisphere showed a decreasing trend of $-3.7\% \pm 1.1\%/10a$ ('a' means year. It is short for 'anniversary') by the end of the 21st century. The rates of SD decline in the TP under different scenarios are different, and the decreasing trend was the most obvious under the RCP8.5 scenario [40–42].

China has a vast territory, and the changes in snow cover among different regions are also different. In the past, most studies on China's snow cover have focused on large-scale single indicators, small-scale multiple indicators, and CMIP5. There are few comprehensive studies on China's snow cover and future changes in CMIP6. Therefore, under the context of global warming, this study selected MODIS and CMIP6 data to analyze the temporal and spatial evolution of snow in the three stable snow cover areas in China. In assessing the recent and future changes in snow cover in China, the objectives were to (1) use MODIS and passive microwave remote sensing data to analyze the main characteristics of the spatial patterns of the SCF, SD, and SCDs in the three stable snow cover areas in China; (2) analyze the main characteristics of the spatial distribution and change trends of the SCF, SD, and SCDs in the future under different scenarios; (3) compare the historical and future changes in the SCF, SD, and SCDs in the three stable snow cover areas. This study can be used as an important reference to provide scientific guidance for formulating the rational utilization of snow resources, future disaster risk assessment, and regional planning in the three stable snow cover areas in China.

## 2. Materials and Methods

### 2.1. Study Area

China is located in the eastern part of Asia. Its terrain is dominated by mountains, plateaus, and hills, and it contains abundant snow resources. Snow cover in China is mainly distributed in the TP, NX, and NC. Known as China's three stable snow cover areas, the total area is about $4.2 \times 10^6$ km$^2$, as shown in Figure 1. The TP is called the "Third Pole" of the Earth, with a mean altitude of more than 4000 m. It is the source of many Asian rivers, so it is also called the "Asian Water Tower" [43]. It has a huge impact on the regional climate, especially the changes in the summer monsoon [44]. NX is located in the northern part of the Tianshan Mountains, which is a typical arid and semi-arid area. It is mainly composed of the Altai Mountains, the Junggar Basin, and the Tianshan Mountains. It is an important economic belt on "The Silk Road" [45]. The shortage of water resources in this area mainly depends on the supply of meltwater from ice and snow. NC includes the three northeastern provinces (Liaoning, Jilin, and Heilongjiang) and the cities of Hulun buir, Hinggan League, Tongliao, and Chifeng in Inner Mongolia. This area is one of China's main food production bases, and changes in snow cover will affect agricultural production activities [46].

### 2.2. Materials

#### 2.2.1. Snow Data

Snow cover data were obtained from MOD10A2-V006 and MYD10A2-V006 provided by the National Snow and Ice Data Center (NSIDC) [47]. MOD10A2 and MYD10A2 are 8-day synthetic snow cover products monitored by the new-generation "Earth Observation System" Terra and Aqua satellites, with a spatial resolution of 500 m. MOD10A2 and MYD10A2 snow cover products are very common in snow studies due to their wide monitoring range and high resolution [9,48]. The products contain two bands. The "Maximum Snow Extent" band represents the maximum snow cover in 8 days; that is, if snow is observed on one day in 8 days, the pixel is marked as snow. If snow is not observed on a single day in 8 days, the pixel is marked as snow-free. The "Eight Day Snow Cover" band represents the number of days of snow cover observed in 8 days; 0 means no snow cover, 1 means snow cover, the 0th digit is the observation result of day 1, and the 7th digit is the observation result of day 8. The results are converted to binary and saved. This

study selected the data of MOD10A2 and MYD10A2 from 1 to 46 periods from 2001 to 2020, and used MODIS Reprojection Tools (MRT) to splice, transform, and project the images of 19 satellite orbital numbers in the study area, and finally output them into TIFF files. The codes of MODIS products were processed, and only snow (code: 200), cloud (code: 50), and lake (code: 37) were retained, and the rest were all snow-free. The MOD10A2 and MYD10A2 products were combined with the maximum value [8] and, combined with the snow depth data, the pixels identified as clouds in the MODIS data were further processed. If it was recognized as a cloud in the MODIS image but the snow depth was greater than 0, it was judged as snow. If the MODIS image was identified as a cloud and the snow depth was also equal to 0, it was judged as a cloud. The missing data of MYD10A2 between 2001 and 2002 were replaced by MOD10A2 data.

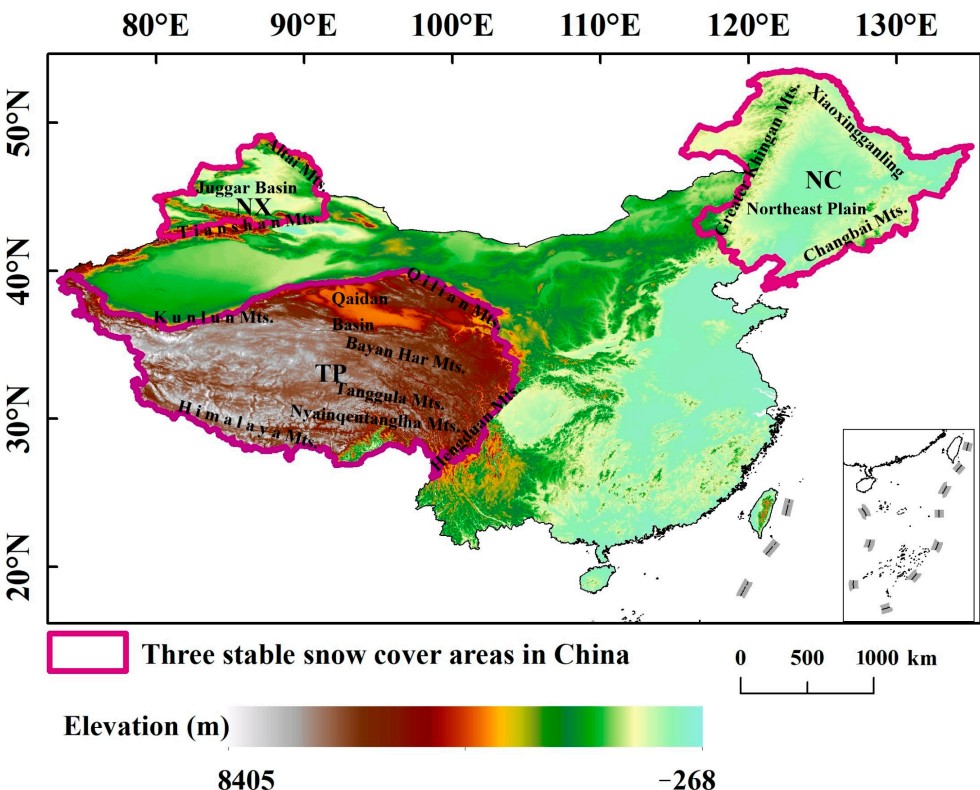

**Figure 1.** Digital elevation model of China and its distribution of three stable snow cover areas in China. The purple frames are the boundaries of the three stable snow cover areas in China. The abbreviations of the snow cover areas represent the Qinghai–Tibet Plateau (TP), North Xinjiang (NX), and Northeast China (NC). The illustration in the lower right corner is the South China Sea, and the dotted line represents the range line of the ownership line of Chinese islands.

### 2.2.2. SD Data

The SD data were collected from the "Long-term series of daily snow depth dataset in China" provided by the National Tibetan Plateau Data Center (TPDC) [49,50]. This dataset used passive microwave remote sensing SMMR, SSM/I, and SSMI/S data to obtain snow depth data in China from 1978 to 2019 through cross-calibration, the determination of snow depth inversion coefficients, and snow cover identification and classification trees. The SD data were saved by the ASCII code. ASCII text data were converted into raster images and resampled to 500 m when calculating the SD.

### 2.2.3. Meteorological Data

The daily precipitation and mean temperature raster data from 1961 to 2019 with a resolution of 0.5° × 0.5° were provided by the National Meteorological Information Center (http://www.nmic.cn, accessed on 6 January 2022). The dataset uses ANUSPLINE

interpolation for 2472 meteorological stations, taking into account the effects of terrain factors and longitude and latitude on precipitation and temperature, and has passed the assessment and verification of the National Meteorological Administration after cross-validation and error analysis. This study converted the monthly precipitation and mean temperature into raster data of spring and winter precipitation and the mean temperature from 2004 to 2008.

### 2.2.4. CMIP6 Data

This study selected the three variables of snow area percentage (snc), snow depth (snd), and mean age of snow (agesno) of 15 modes in CMIP6, as shown in Table 1. These variables include the four future shared socioeconomic pathways (SSPs). They are SSPs126 (sustainable development path), SSPs245 (moderate development path), SSPs370 (unbalanced development path), and SSPs585 (fossil fuel-based development path). After converting these data into the TIFF format, the multi-model mean of each variable was obtained, the units were unified, the data were resampled to 5 km, and the time scale was unified to 2015–2100. Many studies have evaluated and compared the ability of CMIP6 snow cover data to simulate snow cover with that of CMIP5. CMIP6 has more modes than CMIP5 with a correlation greater than 0.9 with historical observations, and CMIP5 is less sensitive to climate [39]. CMIP6 improves the underestimation of snow cover of CMIP5 in some months, and snow cover monitoring in the northern hemisphere is better than in CMIP5 [38,51]. This study used CMIP6 data to study the future changes in snow cover in the three stable snow cover areas in China. By doing a multi-modal ensemble of 15 modes, the limitations of a single mode can be overcome, and the results are more convincing [52].

**Table 1.** The Coupled Model Intercomparison Project (CMIP6) model name, institute, and resolution numbers of each model used in this study. $\sqrt{}$ represents the model used under each variable.

| Model Name | snc | snd | agesno | Institute | Global Grid Resolution |
|---|---|---|---|---|---|
| BCC-CSM2-MR | $\sqrt{}$ | | | BCC | $1.1° \times 1.1°$ |
| CanESM5 | $\sqrt{}$ | $\sqrt{}$ | | CCCMA | $2.8° \times 2.8°$ |
| CAS-ESM2-0 | $\sqrt{}$ | $\sqrt{}$ | | CCCMA | $1.4° \times 1.4°$ |
| CESM2-WACCM | $\sqrt{}$ | $\sqrt{}$ | | NCAR | $1.25° \times 1.0°$ |
| CIESM | $\sqrt{}$ | $\sqrt{}$ | | THU | $1.25° \times 1.0°$ |
| CMCC-ESM2 | $\sqrt{}$ | | | EMCCC | $1.25° \times 1.0°$ |
| EC-Earth3 | $\sqrt{}$ | | | EC-Earth | $0.7° \times 0.7°$ |
| FGOALS-f3-L | $\sqrt{}$ | $\sqrt{}$ | | CAS | $1.25° \times 1.0°$ |
| GFDL-CM4 | $\sqrt{}$ | $\sqrt{}$ | | NOAA-GFDL | $1.25° \times 1.0°$ |
| IPSL-CM6A-LR | $\sqrt{}$ | | $\sqrt{}$ | IPSL | $2.5° \times 1.25°$ |
| MPI-ESM1-2-HR | $\sqrt{}$ | | | MPI-M | $0.9° \times 0.9°$ |
| MRI-ESM2-0 | $\sqrt{}$ | $\sqrt{}$ | | MRI | $1.1° \times 1.1°$ |
| TaiESM1 | $\sqrt{}$ | $\sqrt{}$ | | AS-RCEC | $1.25° \times 1.0°$ |
| KIOST-ESM | | $\sqrt{}$ | | KIOST | $1.9° \times 1.9°$ |
| MIROC6 | $\sqrt{}$ | | | NIES, JAMSTEC | $1.4° \times 1.4°$ |

### 2.3. Methods

### 2.3.1. SCF (Snow Cover Fraction)

SCF [29] represents the proportion of snow cover in the specified area; that is, the percentage of snow cover on the day to the total area of the study area. The calculation formula is as follows:

$$\text{SCF} = \frac{S_{snow}}{S_{all}} \times 100\% \qquad (1)$$

where SCF is the snow cover fraction, $S_{snow}$ is the snow cover area, and $S_{all}$ is the area of the entire study area.

### 2.3.2. SCDs (Snow Cover Days)

SCDs [53] represent the number of times each pixel is covered by snow in a year. The larger the SCD, the longer the snow cover and the more abundant the snow storage is in the area. The calculation formula is as follows:

$$\text{SCD} = \sum_{i=0}^{N} S_i \tag{2}$$

where SCD is the snow cover days; $N = 46$; $S_i$ is the snow cover pixels.

### 2.3.3. Sen's Trend Analysis and Mann–Kendall Significance Test

This study adopted the method of combining Sen trend analysis and the Mann–Kendall significance test to analyze the change trend and significance of snow cover based on the pixel scale. This method has become an important method for judging the trend of time series data and is widely used in meteorology, hydrology, and vegetation time-series change characteristics analysis [54,55]. The Sen trend analysis is obtained by calculating the median value of the series, but it cannot realize the significance judgment of the series trend by itself. The Sen value was tested by the Mann–Kendall test. The Sen trend analysis is calculated as follows:

$$\beta = \text{Median} \left( \frac{x_j - x_i}{j - i} \right) (\forall j > i) \tag{3}$$

where $i$ and $j$ represent the years $i$ and $j$, respectively; $x_i$ and $x_j$ represent the values of years $i$ and $j$, respectively; $\beta$ represents the trend degree; and the $\beta$ value is used to judge the snow cover trend. When $\beta > 0$, the time series shows an upward trend, and vice versa.

For the sequence $X = (x_1, x_2, \ldots, x_n)$, first determine the magnitude relationship between $x_i$ and $x_j$ in all dual values $(x_i, x_j, j > i)$ and define the statistics of sequence S:

$$S = \sum_{1=1}^{n-1} \sum_{j=i+1}^{n} \text{sgn}(x_j - x_i) \tag{4}$$

$$\text{sgn}(x_j - x_i) = \begin{cases} +1 \ (x_j - x_i > 0) \\ 0 \ (x_j - x_i = 0) \\ -1 \ (x_j - x_i < 0) \end{cases} \tag{5}$$

Then calculate the variance of S:

$$\text{Var(S)} = \frac{n(n-1)(2n+5)}{18} \tag{6}$$

Finally, S is transformed into the statistics of Z, and the test statistic Z is calculated by

$$Z = \begin{cases} \frac{S-1}{\sqrt{\text{Var(S)}}} (S > 0) \\ 0 \ (S = 0) \\ \frac{S+1}{\sqrt{\text{Var(S)}}} (S < 0) \end{cases} \tag{7}$$

where S is the test statistic; $x_i$ and $x_j$ represent the values of years $i$ and $j$, respectively; n is the number of samples in the sequence; $\text{sgn}(x_j - x_i)$ is a symbolic function; Var(S) is the variance of S; and Z is the standardized test statistic. When $|Z| \geq Z_{(1-\alpha/2)}$, the trend is significant. The significant level selected in this test was $\alpha = 0.05$, $Z_{(1-\alpha/2)} = 1.96$. If the pixel point passes the significance test, the change trend was analyzed according to the value of the trend degree $\beta$. The Mann–Kendall significance test is described in detail by Daufresne et al. [56].

## 3. Results

### 3.1. Variation Characteristics of Snow Cover since the 21st Century

Figure 2 is an analysis of the interannual variation in the SCF (a), SD (b), and SCDs (c) in the three stable snow cover areas. The mean values of the SCF, SD, and SCDs in NX

were the highest, which were 37%, 3.43 cm, and 47.81 days, respectively. The TP had a greater SCF value than NC, a lower SD value than NC, and a lower SCD value than NC after 2009. In 2008, affected by the snow disaster in southern China caused by La Niña, the TP's SCF, SD, and SCD values all improved [57], and the SCF of that year surpassed that of NX. There was no significant change trend in the three stable snow cover areas ($p > 0.05$). From 2004 to 2010, the change trends of NX and NC were roughly the same. After 2008, the change trend of the SCF of NC was opposite to that of the TP. The change trends of the SD in the three stable snow cover areas all showed increasing trends. The SD of NX and NC was high and fluctuated greatly, while the SD and SCD values of the TP were low but the change was relatively stable.

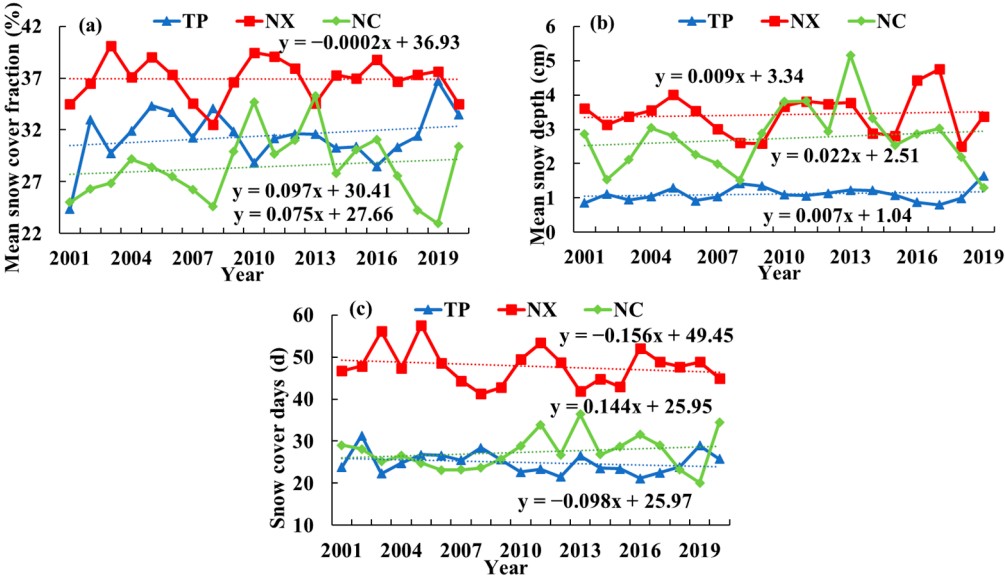

**Figure 2.** Annual variation in SCF (**a**), SD (**b**), and SCDs (**c**) in the TP, NX, and NC from 2001 to 2019 or 2020. The blue line represents the TP, the red line represents NX, and the green line represents NC.

Figure 3 shows the seasonal variation characteristics of the SCF (a) and SD (b) in the three stable snow cover areas in China. The SCF and SD values of the TP were higher than those of the other two snow cover regions in summer and autumn, and the seasonal fluctuation was the smallest, indicating that the stability of the TP snow cover is the best. However, the snow cover of NX and NC is mainly concentrated in spring and winter, and the snow cover in summer is very low, indicating that the snow cover of NX and NC is mainly seasonal snow cover, mainly relying on snowfall supply in high-latitude winter. The changes in the SCF and SD values in the four seasons of NC showed increasing trends; the SCF increased most in autumn, with a change rate of 1.95%/10a. The changes in the SCF and SD values of NX in spring showed a decreasing trend, with rates of change of −2.09%/10a and −0.03 cm/10a, respectively. The SD of the TP showed an increasing trend except in autumn.

Combining Figures 2 and 3, it can be seen that NX showed an obvious decreasing trend from 2005 to 2008, which was related to the decrease in snow cover in spring and winter. The SD of NC showed a significant decreasing trend after 2013, with a rate of change of −4.76 cm/10a ($p < 0.05$), especially in spring and winter. The rate of change in spring was −6.78 cm/10a, and the rate of change in winter was −12.3 cm/10a. Spring and winter happen to be the snowiest periods in the areas. Against the background of global warming, and affected by the mid-latitude westerly circulation and the dry and cold arctic airflow; although, the amount of snowfall is increasing, the frequency of snowfall is decreasing, resulting in a decrease in snow cover [9,58].

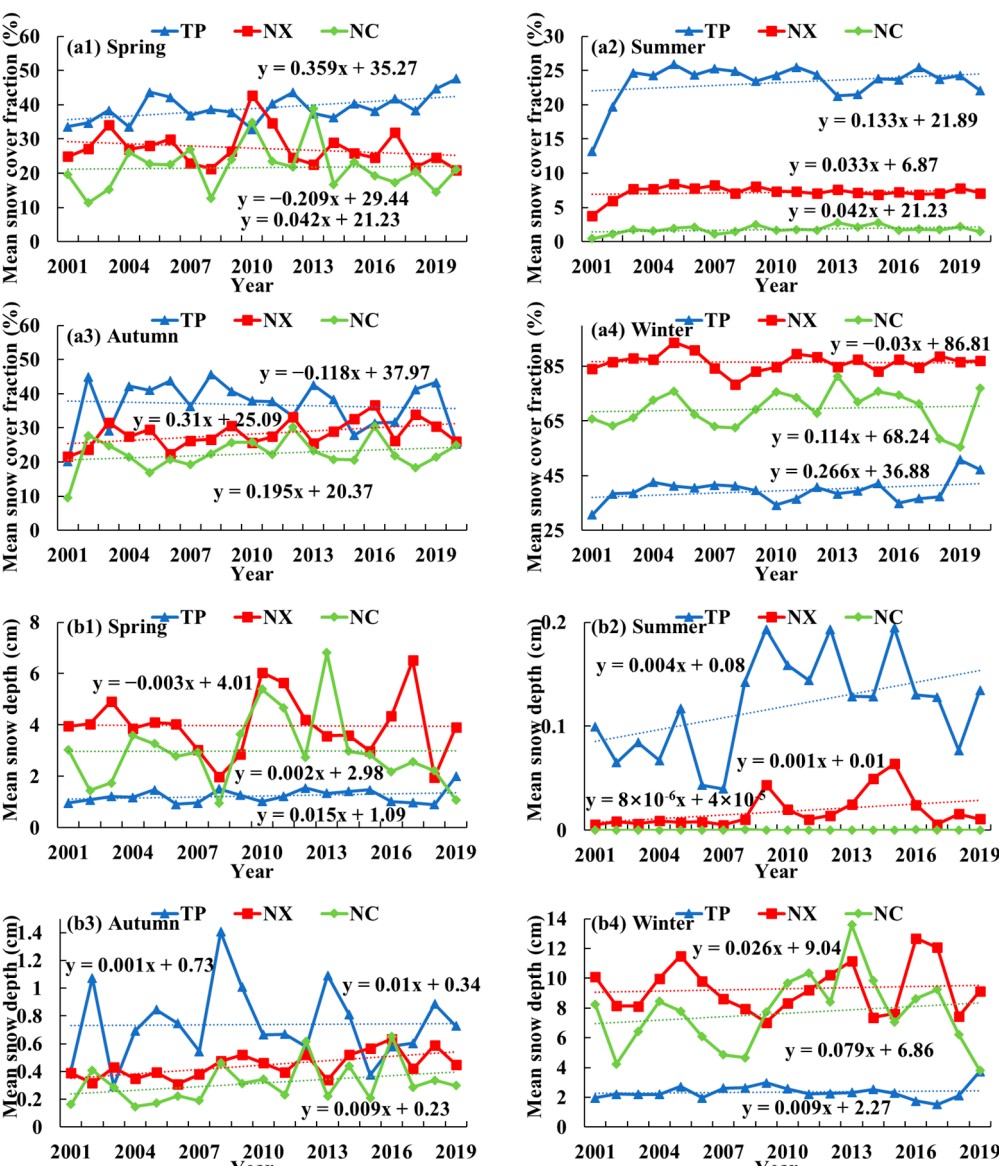

**Figure 3.** Annual variations in the SCF (**a**) and SD (**b**) in spring (**a1,b1**), summer (**a2,b2**), autumn (**a3,b3**), and winter (**a4,b4**) in the TP, NX, and NC from 2001 to 2019 or 2020. The blue line represents the TP, the red line represents NX, and the green line represents NC.

Figure 4 shows the interannual variations in the SCF and snow cover areas (a) and the SD and SCDs (b) at different elevations in the three stable snow cover areas. The three indicators are positively correlated with the elevation. The three indicators in the TP at >6000 m are much higher than the snow cover below 6000 m, and the three indicators below 5000 m are significantly lower than the snow cover at the same altitude in NX and NC. The three indicators of NX are generally larger than that of the TP and NC at the same elevation, while the snow areas of the TP are larger than that of NX and NC at high altitudes. Snow cover in NC did not differ much at different elevations, suggesting that the region is less affected by elevation. The snow cover in the three stable snow cover areas has obvious changes in the elevation >2000 m, the SCF and SD generally increase in the area with elevation >2000 m, and the SCDs tend to decrease. There is no obvious change trend in areas with altitudes below 2000 m. In areas of the TP >6000 m, the SD showed a significant increasing trend, with a change rate of 1.3 cm/10a; the SCDs decreased significantly, with a change rate of −10.2 day/10a. The areas >6000 m are mainly distributed in the Kunlun Mountains in the west of the TP and the Himalayas in the south. Under the influence of

the westerly wind and the southwest monsoon, the cold air mass in the northern part is weaker in spring and autumn, and the cold air mass in the winter is stronger. The area is located in a monsoon climate zone, with sufficient water vapor in the south, and the cold air from the north is prone to snowfall, leading to an increase in the SD; however, when the temperature is higher, the snow will melt rapidly, resulting in a decrease in SCDs [59,60].

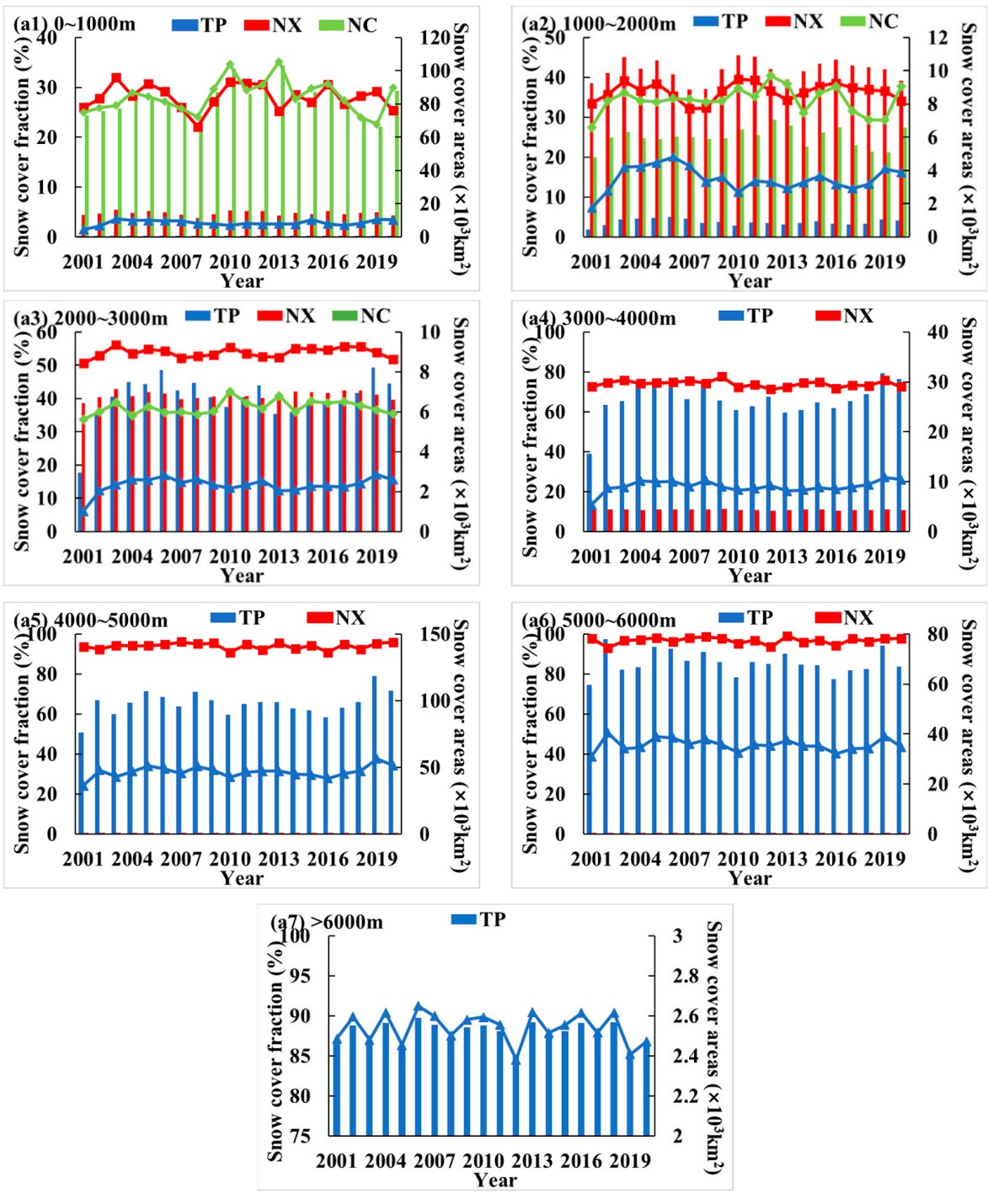

**Figure 4.** *Cont.*

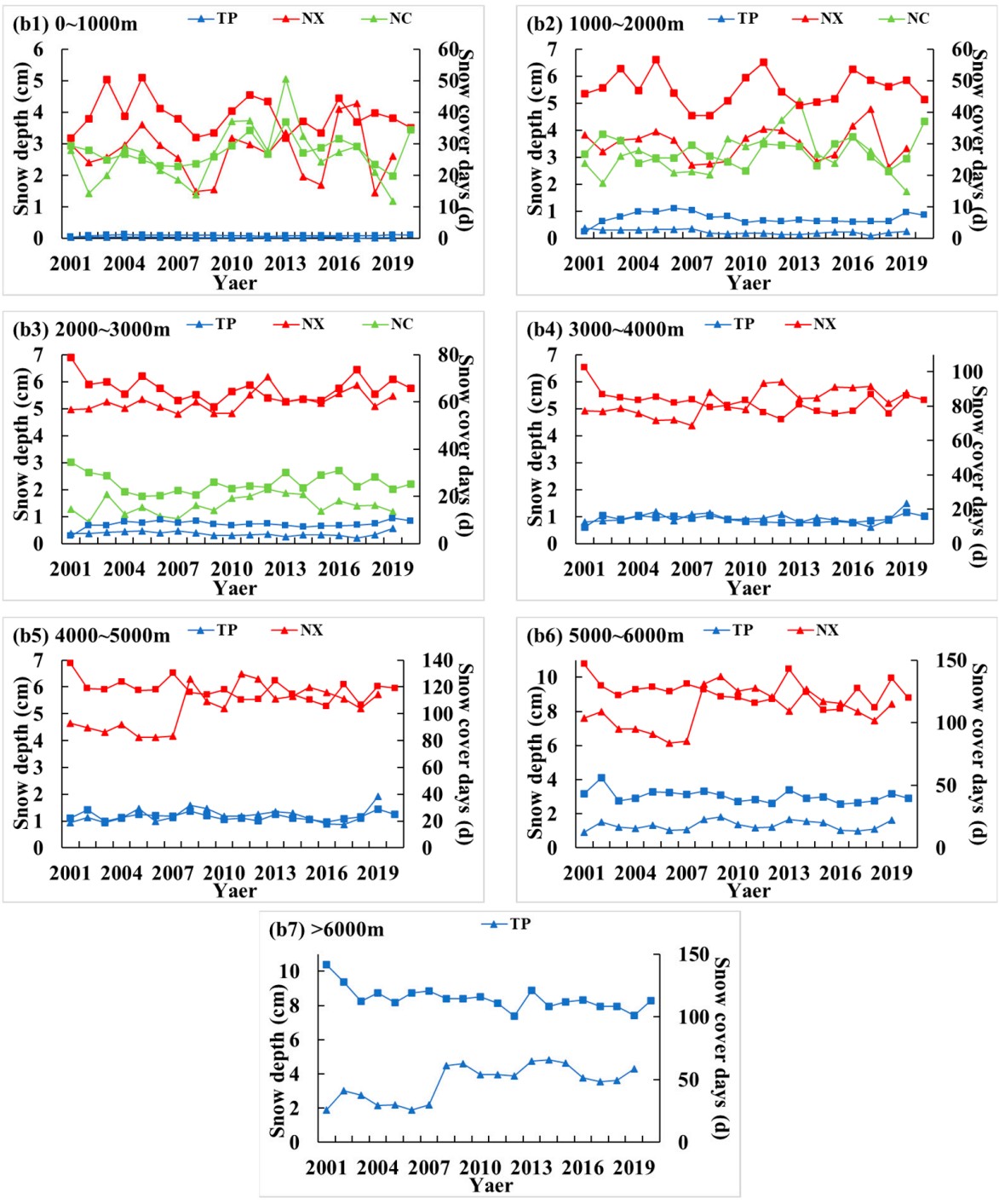

**Figure 4.** Annual variations in the SCF and snow cover areas (**a**) and SD and SCDs (**b**) at various altitudes in the TP, NX, and NC from 2001 to 2019 or 2020. Blue represents the TP, red represents NX, and green represents NC. In (**a**), the broken line represents the SCF, and the column represents snow cover areas. In (**b**), the triangles in the broken line represent the SD and the squares represent the SCDs.

Figure 5 shows the spatial distribution characteristics of the SCF (a), SD (b), and SCDs (c) in the three stable snow cover areas, and the three indicators are consistent in spatial distribution. The high-value areas of snow cover are mainly distributed in the southeastern Nyainqentanglha Mountains and the western Kunlun Mountains in the TP, the southern Tianshan Mountains and the northeastern Altai Mountains in NX, and the northern Daxinganling and Xiaoxinganling in NC. The snow cover of the TP has strong

spatial heterogeneity and is blocked by tall mountains. There is a lot of snow in the surrounding areas and little in the central hinterland. The Qaidam Basin and the northern Tibetan Plateau are areas with less snow [61]. NX is located in a temperate continental climate zone, with a large inter-annual temperature difference, high summer temperatures, and high evaporation, which is easily affected by drought, and snow does not easily accumulate. Snow cover is affected by altitude and latitude, and the Junggar Basin with low altitude is an area with less snow. However, the high-altitude Tianshan Mountains and the Altai Mountains are affected by the westerly winds, forming snowy areas in the interception of vegetation [62,63]. The latitudinal zonality of NC is obvious. There are important heavy industry bases in China, and human activities are the highest in the three areas. The reason for the low snow cover in the south–northeast plain may be related to human activities. The north of NC is the coldest area in China, and it is located in a semi-humid area with a lot of snowfall, and the snow cover is widely distributed in low-altitude areas. The SCFs of the three stable snow cover areas are mainly concentrated between 20 and 40%, at 36.6% (TP), 69.8% (NX), and 55% (NC), respectively. Among the three stable snow cover areas, the TP has the highest proportion in the area with an SCF greater than 60%, accounting for 11.4%, while the area with an SCF greater than 60% in NC accounts for only 0.01%. Areas with high SD have lower temperatures, higher precipitation than surrounding areas, lower solar radiation, and shorter sunshine hours, which provide favorable conditions for the long-term preservation and accumulation of snow. NX and NC have high latitudes and abundant snowfall in winter. The areas with SD values above 4 cm account for 27.6% and 27.1%, respectively, which are greater than the 4.5% for the TP. There are mainly between 1 and 30 SCDs in the TP and NC, and the area ratios are 72.9% and 62.7%, respectively. There are between 30 and 60 SCDs in NX, and the area ratio is 48.4%. Areas with more than 60 SCDs are called stable snow cover areas [14], and the proportion of NX stable snow cover is 26.2%, which is greater than the 9.9% and 9.5% of the respective TP and NC.

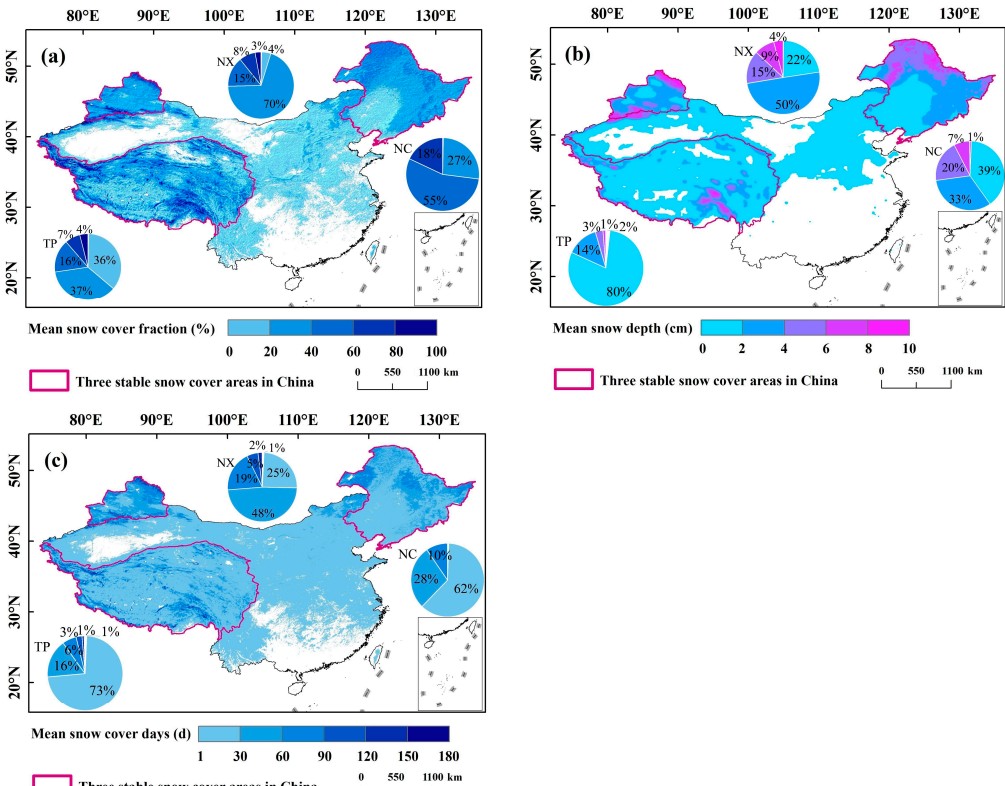

**Figure 5.** Spatial distribution of average annual SCF (**a**), SD (**b**), and SCDs (**c**) in the TP, NX, and NC from 2001 to 2019 or 2020. The pie chart represents the percentage of the total number of different intervals, and the color corresponds to the color of the legend.

Figure 6 shows the results of the Sen trend analysis and Mann–Kendall significance test of the SCF (a), SD (b), and SCDs (c) in the three stable snow cover areas. The three stable snow cover areas are quite different in internal space, but the changes in the three indicators also show similarities.

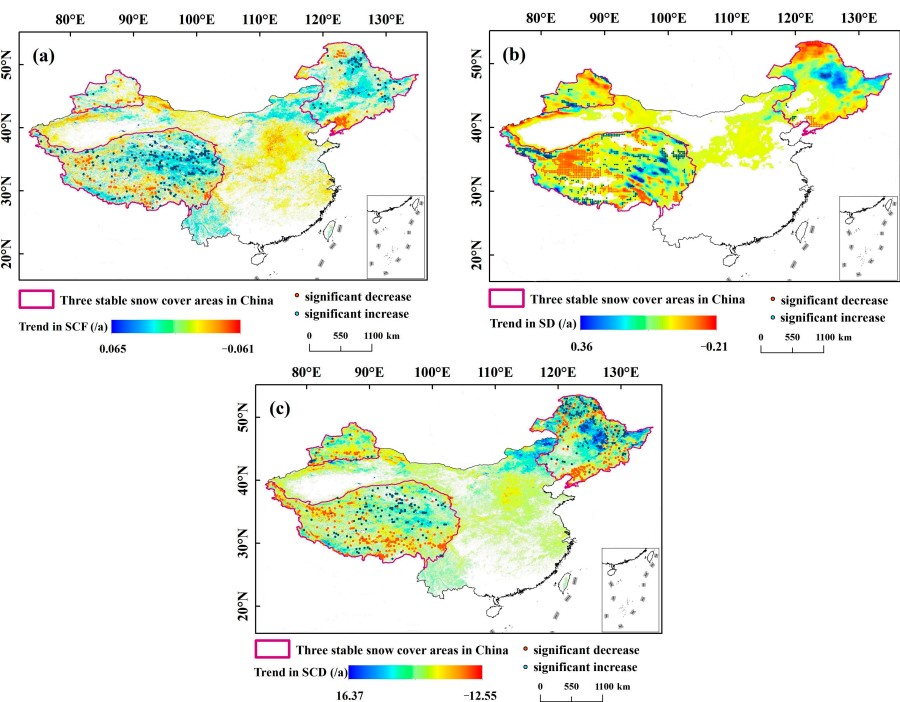

**Figure 6.** Spatial distribution of variation ratios and the significance of SCF (**a**), SD (**b**), and SCDs (**c**) in the TP, NX, and NC from 2001 to 2019 or 2020.

The TP snow cover increased in the area between the Bayankala Mountains and Qilian Mountains in the northeast and the Himalayas in the southwest, and decreased in the Nianqing Tanggula Mountains and Hengduan Mountains in the southeast and the Northern Tibetan Plateau in the middle. The change in NX snow showed a change trend from southwest to northeast. The snow in the eastern Tianshan and Altai Mountains showed an increasing trend, and the central Junggar Basin mainly showed a decreasing trend. NC showed an increasing trend in the area between the Daxinganling and the Xiaoxinganling in the central part, and a decreasing trend in the southern Liaodong Peninsula and the northern part of the Daxinganling (SCF, SD) in the north. Among the three stable snow cover areas, the area with no change in the SCF accounted for the largest proportion, and the area where the SCF of the TP and NC showed an increasing trend was larger than the area that showed a decreasing trend. Areas with significant changes in the SCF showed smaller differences, with the largest differences in the TP, with 3% more areas with significant increases than those with significant decreases.

The SCF of the TP and NC mainly increased in the region by 20–40%, 46.4%, and 43.5%, respectively. The TP and NX showed a decreasing trend in areas with an SCF of >60%, which were 37.5% and 27.4%, respectively. The SD of the TP and NC accounted for the largest area with a decreasing trend, accounting for 51.3% and 58%, respectively, while NX showed an insignificant decreasing trend, accounting for 52% of the area. The area with a significant change in the TP accounted for the largest proportion, accounting for 29.9%; the area with a significant change in NX had the largest difference, and the area with a significant increase was 5.6% more than the area with a significant decrease. The areas with an SD of >4 cm in the TP and NC showed an increasing trend, accounting for 75.2% and 55.5%, respectively, while NX was the opposite.

The SCDs of the TP and NX mainly had a decreasing trend, accounting for 42.5% and 57.6% of the area, respectively. The NC showed the largest area of SCDs with a significant

increase and a significant decrease, which were 6.6% and 8.3%, respectively. The stable snow cover of the TP and NX mainly had a decreasing trend, accounting for 78.1% and 49.2%, respectively, and in NC, it mainly had an increasing trend, accounting for 78.2%.

### 3.2. Variation Characteristics of Snow Cover in the Future Based on CMIP6

Figure 7 shows the spatial distribution of bias between CMIP6 and MODIS data in the TP, NX, and NC. In the three stable snow cover areas, NX had the smallest deviation and the TP had the largest deviation. The bias of CMIP6 was smaller at low altitudes, in the northeast of NC, and in the middle of NX, and larger at high altitudes of the TP. The overall bias of the SCF was the smallest, with an average deviation of 74–77% in the TP, 41.4–45% in NX, and 87% in NC. However, there is still room for improvement in the simulation of snow depth. The reason for the large deviation is that the snow-rich areas are high in altitude, which is not conducive to snow monitoring. Due to the difference between the driving factors and spatial resolution of the CMIP6 model, there is an error between the model results and the actual results [64].

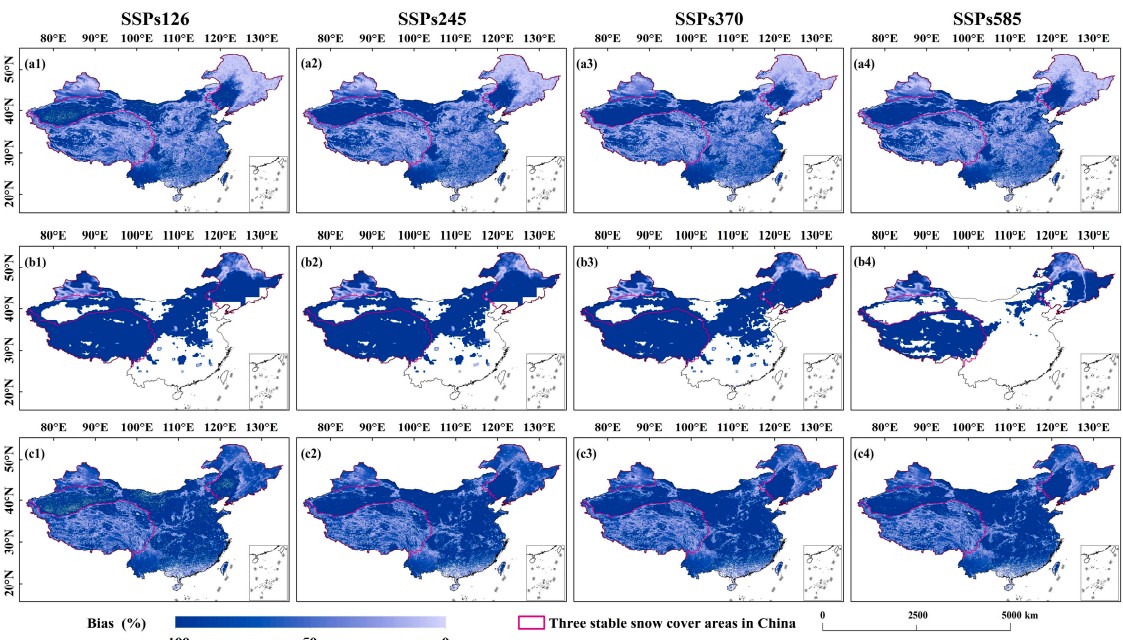

**Figure 7.** Spatial distribution of bias between CMIP6 and SCF (**a**), SD (**b**) and SCD (**c**) in the TP, NX, and NC under the SSPs126 (**a1–c1**), SSPs245(**a2–c2**), SSPs370 (**a3–c3**) and SSPs585 (**a4–c4**) from 2015 to 2019 or 2020.

Figure 8 shows the future interannual variations in the SCF (a), SD (b), and SCDs (c) in the three stable snow cover areas.

Under the SSPs126 scenario, the SCF, SD, and SCDs of the three stable snow cover areas showed a trend that first decreased and then increased, and the decreasing trend was mainly concentrated between 2015 and 2035. The TP changed the fastest in the SD and SCDs, decreasing by 14.2% and 5.6%, respectively, by 2035. NC had the most significant reduction in the SCF, with a 22.7% reduction by 2050. By 2100, the SD showed a slight increasing trend compared to 2050, increasing by 9.4% (TP), 8.2% (NX), and 5% (NC), respectively.

Under the SSPs245 scenario, the SCF, SD, and SCDs in the three stable snow cover regions showed a more obvious decreasing trend than that of the SSPs126. By 2050, TP changed fastest in the SCF and SD, decreasing by 15.3% and 33.3%, respectively. After 2050, there were differences in the changes of the SCDs in the three stable snow cover areas, and the SCDs of NC maintained a downward trend. By 2100, the SCDs decreased by 1.91% compared to 2050. The SCDs of the TP and NX showed a slight increasing trend, and the SCDs in 2100 increased by 0.3% and 3.2% compared to 2050.

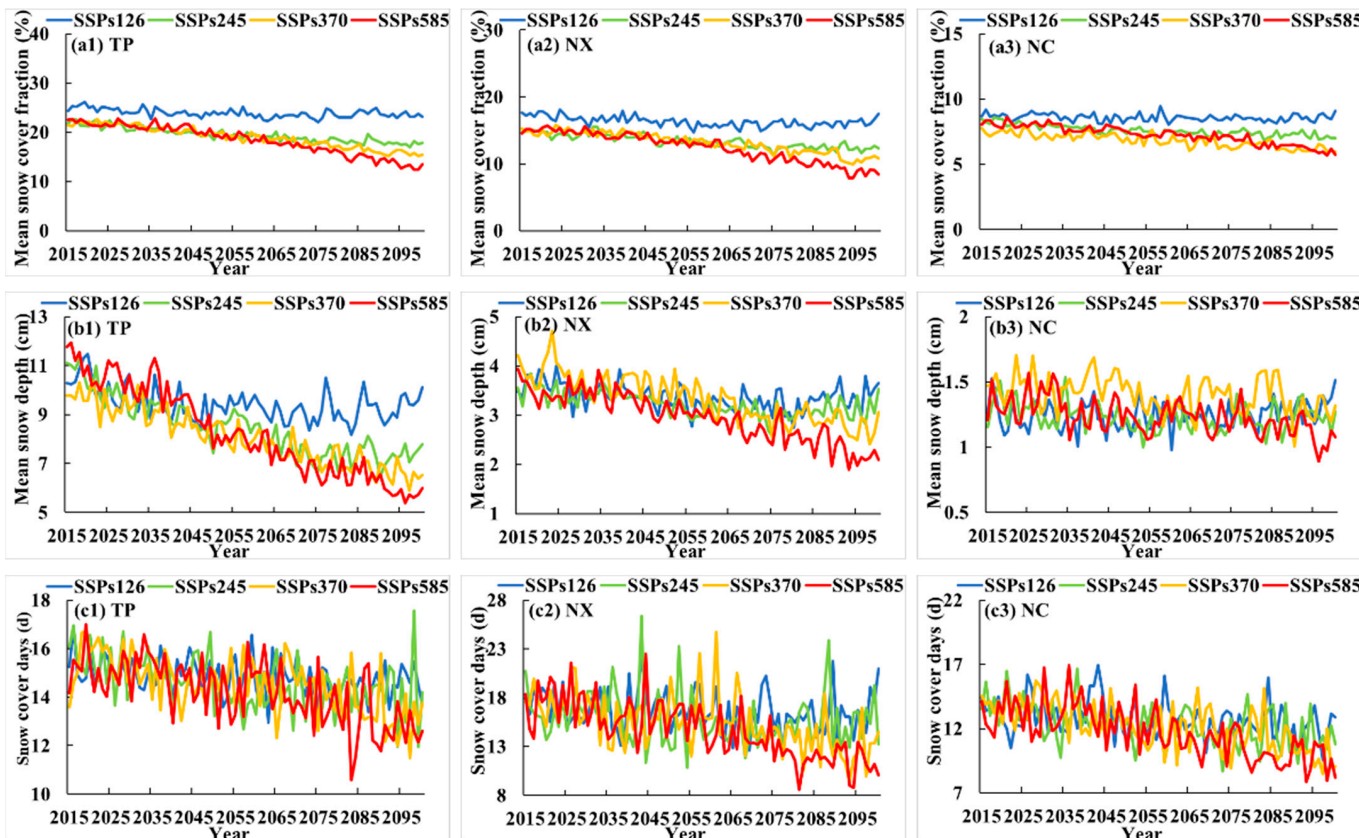

**Figure 8.** Annual variations in the SCF (**a**), SD (**b**), SCDs (**c**) in the TP (**a1**–**c1**), NX (**a2**–**c2**), and NC (**a3**–**c3**) under SSP scenarios from 2015 to 2100. The blue line represents SSPs126, the green line represents SSPs245, the yellow line represents SSPs370, and the red line represents SSPs585.

Under the SSPs370 scenario, the SCF, SD, and SCDs of the three stable snow cover regions showed a downward trend in continuous fluctuations from 2015 to 2100. NX had the fastest changes in the SCF and SD, decreasing by 9.3% and 15.6% by 2050, and by 29.3% and 33.2% by 2100, respectively. NC's SCF varied the most around 2050.

Under the SSPs585 scenario, the decreasing trends of the SCF, SD, and SCDs in the three stable snow cover areas will continue to increase over time. NX changed the fastest in the SD and SCD, especially with the reduction in the SD, decreasing by 16.5% by 2050, 39.1% by 2075, and 46.6% by 2100. The largest reduction in the three stable snow cover areas was the SCF, which decreased by more than 40% by 2100.

As the development imbalance increased, the snow cover reduction rate of NX was the fastest among the three stable snow cover areas, while the TP had the smallest change in snow cover under different scenarios.

Figure 9 shows the difference between 2015–2050 and 2051–2100 in the average SCF (a), SD (b), and SCDs (c) of the three stable snow cover areas. In the second half of the 21st century, compared to the first half of the 21st century, the areas where the SCF decreased by more than 3% were mainly distributed in the eastern part of the TP under the SSPs245 scenario, and the reduced area continued to expand westward under the SSPs370 scenario. Under the SSPs585 scenario, most of the TP except the southern Himalayas, the southwest of NX, and the northern part of NC were also significantly reduced. The decrease in the SD in the second half of the 21st century was mainly distributed in the west and southeast of TP. The decreases in SSPs245, SSPs370, and SSPs585 were more than 3 cm, and the decrease in the Karakoram Mountains under SSPs585 was the largest, at 19.7 cm. The reductions in the SCDs of NX and NC under different scenarios were all less than 1 day. In the second half of the 21st century, the SCDs decreased most significantly in the TP. Similar to the SD, the west and southeast were at the center of the decrease in SCDs, and

the Nyainchen Tanggula Mountains decreased by 2.5 days, but there were areas of increase in the southwest. Under the SSPs585 scenario, the Junggar Basin in central NX and the northeastern plain in southern NC were also reduction centers of SCDs.

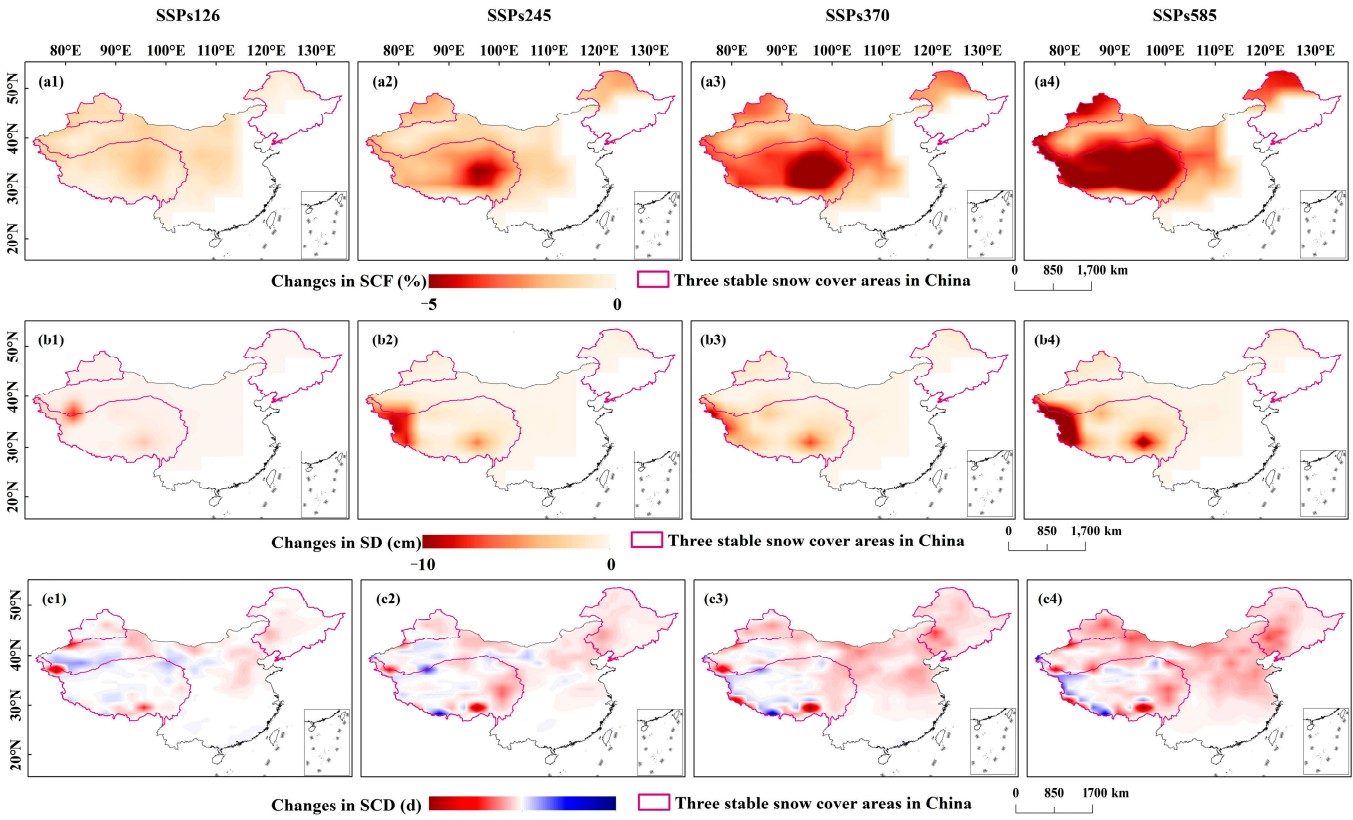

**Figure 9.** Spatial distribution of average SCF (**a**), SD (**b**), and SCDs (**c**) in the TP, NX, and NC under SSPs126 (**a1–c1**), SSPs245 (**a2–c2**), SSPs370 (**a3–c3**), and SSPs585 (**a4–c4**) between 2015–2050 and 2051–2100.

Figure 10 shows the results of the Sen trend analysis and the MK significance test of the SCF (a), SD (b), and SCDs (c) of the three stable snow cover areas under different future scenarios. The future snow cover decreased most significantly in the southeast of the TP, the northwest of NX, and the north of NC. The SD also showed a significant decreasing trend in the west of the TP, but the SCDs had an increasing trend in this area.

As the development imbalance increased, the SCF of the TP decreased the most, and the rates of change were the most significant under the four scenarios, which were −0.18%/10a, −0.52%/10a, −0.84%/10a, and −1.15%/10a, respectively. There was a further downward trend in the Karakoram Mountains, Gangdise Mountains, and Kunlun Mountains in the north of the TP. In the future, the downward trend of NX was smaller than that of the TP, mainly concentrated in the western region.

The SD of the TP decreased most significantly, by −0.11 cm/10a, −0.41 cm/10a, −0.41 cm/10a, and −0.69 cm/10a, respectively, mainly distributed in the western Karakoram Mountains and the southeastern Nyainqentanglha Mountains. Except for the SSPs126 scenario, the areas with significantly reduced SD in the TP and NX accounted for more than 90%, and the area with a significantly decreased SD in NC was less than that of the other two snow areas, accounting for 21.28%, 25.24%, and 37.37% respectively, and mainly distributed in the north of Daxinganling.

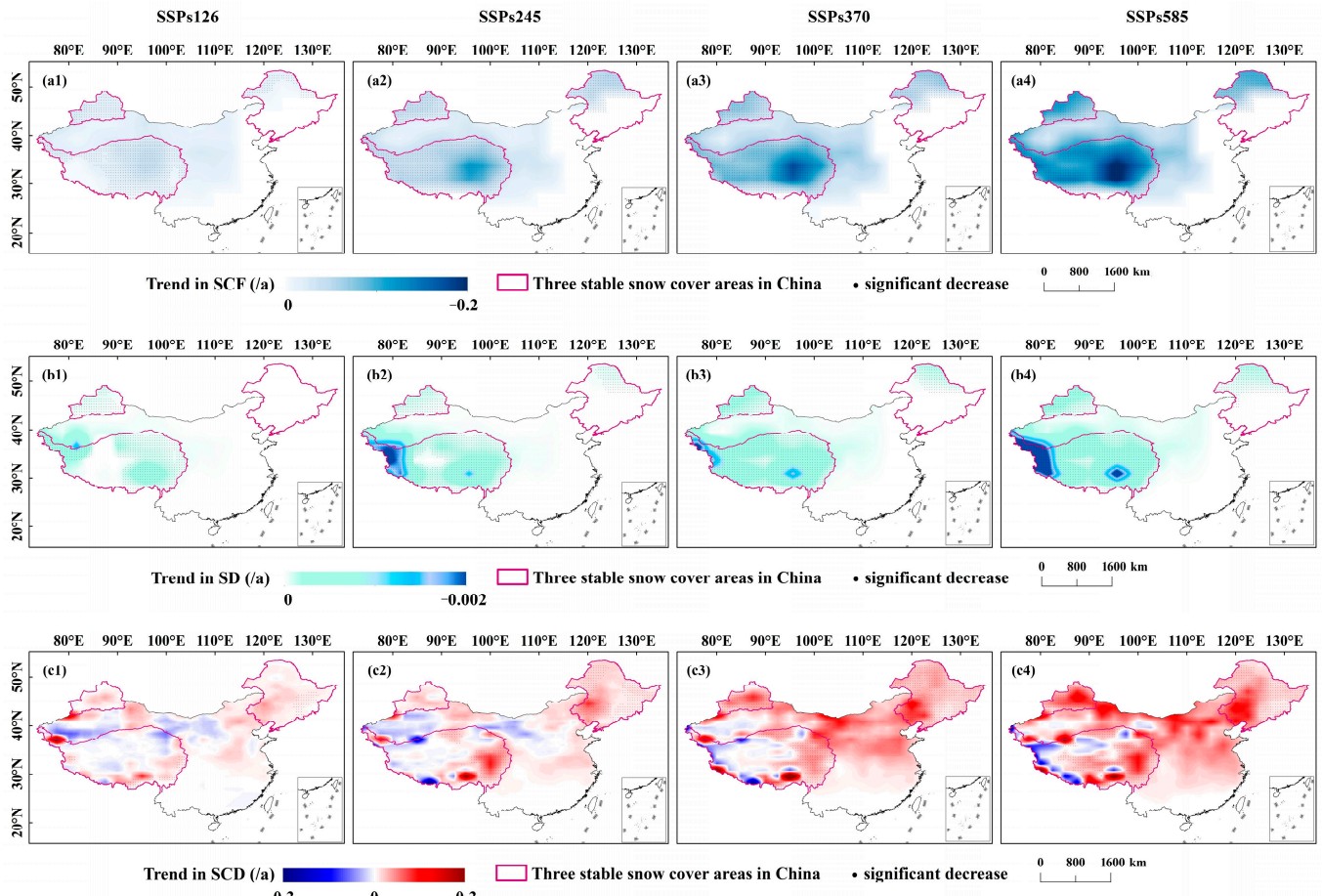

**Figure 10.** Spatial distribution of variation ratios and the significance of SCF (**a**), SD (**b**), and SCDs (**c**) in the TP, NX, and NC under SSPs126 (**a1–c1**), SSPs245 (**a2–c2**), SSPs370 (**a3–c3**), and SSPs585 (**a4–c4**) from 2015 to 2100.

The SCDs of NX decreased most significantly in different scenarios, by −0.16 day/10a, −0.19 day/10a, −0.56day/10a, and −0.86 day/10a, respectively. Under the SSPs245 and SSPs370 scenarios, the area where NX showed a significant decrease was significantly different between the two scenarios, with a difference of 75.6%. In NC, except for SSPs126, the proportion of the area with a significant decrease in SCDs was greater than 80%. The area with a significant decrease in the TP accounted for the smallest proportion and was mainly distributed in the east of the Qaidam Basin and Hengduan Mountains.

## 4. Discussion

Based on MOD10A2 and MYD10A2 snow cover data combined with passive microwave remote sensing snow depth data, this study comprehensively analyzed the changes in the SCF, SD, and SCDs in China's three stable snow cover areas from 2001 to 2020. From 2001 to 2020, NX had the highest average SCF, SD, and SCDs. In the context of global warming, there was no significant change in the three indicators in the three snow cover areas, which may be related to the intermittent period of warming [65]. The southeast and south of the TP are areas with high SCD and SD values, and the spatial heterogeneity of snow is strong. The external tall mountains are conducive to the accumulation of snow, but due to the barrier of tall mountains, there is less snow in the hinterland of the TP [61]. On the whole, there was no obvious change in the three snow indicators of the TP, but the snow change in the eastern part of the TP was large, which is one of the most significant areas of snow change in Eurasia [66]. Under the influence of summer monsoon rainfall, this area showed a warming and drying trend, which is different from the overall warming

and wetting trend of the TP [67]. The eastern part of the TP showed a warming trend [68]. In the future, the Bayan Har Mountains of the TP will be the center of temperature rise [69], which will lead to a reduction in snow cover in this area and it will become the center of reduction in the TP in the future. Under the SSPs585 scenario, the increase in temperature in the TP and NX was higher than that of other regions, especially in the second half of the 21st century. The temperature rise was larger than the average in China, indicating that arid and semi-arid regions are particularly sensitive to future climate warming [70]. Under the SSPs585 scenario, the temperature rise in winter was higher than that in summer, and the most warming occurred in the northeast, northwest, and TP regions of China at high latitudes and high altitudes [71]. Future temperature increases led to significant reductions in snow cover in the future in TP and NX. Changes in glaciers were the same as those in snow cover, with rapid glacier shrinkage in the southeastern TP [72,73]. Relevant studies have shown that the TP snow cover is regarded as a regulator that may affect the extreme climate in China [74], and its changes are related to the high temperatures and heatwaves in summer in China [75]. Snow cover in NX was mainly concentrated in the high-altitude mountains, and the central hinterland was mostly seasonal snow. The Xinjiang region as a whole showed a trend of warming and humidification, especially in the mountainous areas [76], which also increases the uncertainty of the future changes in snow cover in Xinjiang. The shrinkage of glaciers in the region is also increasing, especially in the Tianshan Mountains [77]. Snow cover in NC is more closely related to latitude, and the south is vulnerable to human activities. From 2004 to 2008, the three snow indicators of NX and NC all decreased, mainly in spring and winter, which may be related to the changes in temperature and precipitation in this area. Figure 11 shows a trend of increasing temperatures in spring and a decreasing trend of precipitation in winter between 2004 and 2008 in these two snow cover areas with the statistics of meteorological data.

When using remote sensing images to study snow, the image is very important for the accuracy of snow recognition. The accuracy rate of MOD10A2 measured in the TP was between 84% and 91%, and the accuracy was positively correlated with the SCDs [78]; the accuracy rate was 83% under clear weather in NX [34]. The cloud covers of MOD10A1 and MYD10A1 in NC were 50% and 53%, respectively [79]. It is very important to remove the interference of cloud pixels for snow research, which affects the accuracy of observation results. Many studies have also conducted research in this area. The current cloud removal method mainly uses the combination of MOD10A1, MYD10A1, and 8-day products MOD10A2 and MYD10A2, and uses the Terra and the Aqua to remove the cloud from the observation results of the same area at different time periods [9,80]. Using this method, the cloud cover of the TP was reduced to 25.2% [81], the total accuracy of the snow cover measured in Xinjiang was 94.2%, and the cloud cover was reduced by 45.99% in Northeast China [82]. The advantage of passive microwave remote sensing is that it will not be affected by cloud pixels, but it also has limitations in research due to its high resolution [83]. In this study, a multi-source remote sensing cloud removal method combining MOD10A2 and MYD10A2 snow data and passive microwave remote sensing data was used, which further eliminated the interference of cloud pixels and improved the overall snow recognition rate. There are also shortcomings in the study of the remote sensing of snow cover. Compared to the meteorological station, the time for using remote sensing to monitor snow cover is shorter, and it is impossible to obtain long-term snow cover data.

The general circulation model (GCM) is reliable at large scales and uncertain at regional scales. Sources of uncertainty include future emission scenarios, small regional climate change, and model uncertainty [84]. The bias of the GCM itself also affects the reliability of the regional climate model (RCM) [85]. The bias correction based on the CMIP6 multimodal mean does not obtain good results in terms of snow depth. Related studies also show that CMIP6 overestimates the SWE in the northern hemisphere [86].

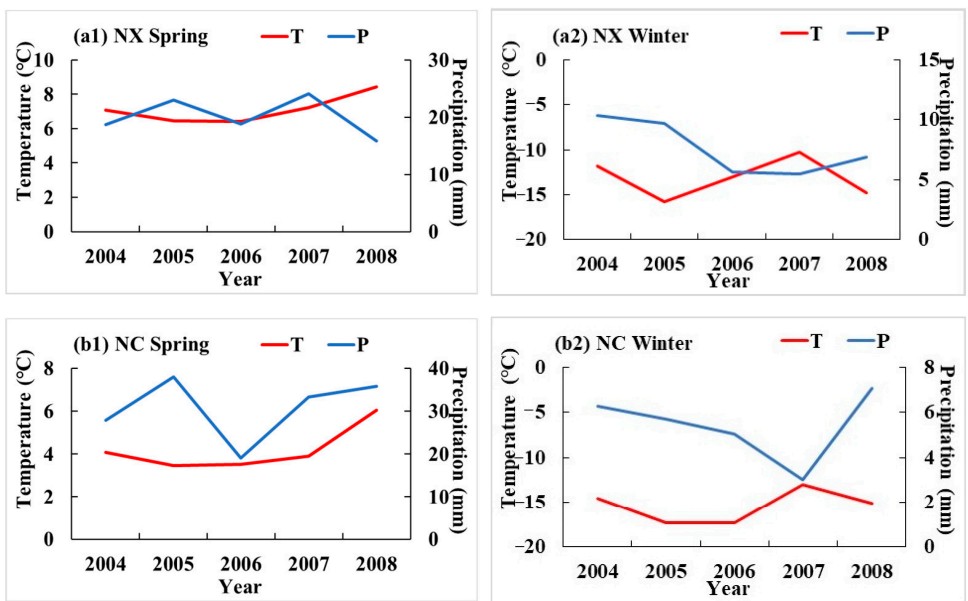

**Figure 11.** Annual variation between the temperature and precipitation of NX (**a1,a2**) and NC (**b1,b2**) from 2004 to 2008. The red line represents temperature and the blue line represents precipitation.

The results of CMIP6 show that the SCF, SD, and SCDs of the three stable snow cover areas will mainly decrease under the four different scenarios in the future, and the decreasing trend was the most significant under the SSPs585 scenario. Under the SSPs126 scenario, the snow cover showed an increasing trend in the second half of the 21st century. The change rate of the SCF was between $-1.15\%$ and $-0.01/10a$, the change rate of the SD was $-0.69$ to $-0.004\,cm/10a$, and the change rate of SCDs was $-0.86$ to $0.05\,day/10a$. Many studies have used CMIP5 to study future changes in snow cover. In the 21st century, the seasonal SCF of ice-free land in the northern hemisphere showed a decreasing trend, and the reduction range was between 7.2% and 24.7% [40]. The rate of change in the future SD of the TP under the four scenarios of CMIP5 was between $-1.1$ and $-0.8\,cm/10a$ [42]. Under the RCP4.5 and RCP8.5 scenarios, the SCDs showed downward trends in the middle and end of the 21st century. The SCD reduction under the RCP8.5 scenario was much greater than that under the RCP4.5 scenario, and the TP was the central area of the reduction [87]. The reduction in the SCF in the northern hemisphere in the next 80 years, especially in the TP, is closely related to the amount of emissions [39].

It is concluded that the reduction in snow cover in CMIP5 was generally lower than that in the results of CMIP6 in this study, which may be related to the consideration of shared socioeconomic pathways (SSPs) and the response of simulated snow cover to solar radiation in CMIP6 scenarios. The future TP glacier area will be reduced to 32–64% of the current area under different RCP scenarios [88], and the glacier reduction rate will be greater than the future snow cover reduction rate in this study. Snow cover is the most sensitive response factor to environmental changes in the cryosphere, and its future changes are closely related to changes in temperature and precipitation. Under the premise of only considering precipitation changes, PRCPTOT, RX5day, and R95P in the northwest region showed increasing trends in the future, and CDD showed a significant decreasing trend [89]. However, after comprehensive consideration of temperature and precipitation, the humidification trend in Northwest China will reverse in the future with the high growth in temperature and the low growth in precipitation [90]. It can be seen that the future temperature increase is one of the important factors leading to the reduction in snow cover.

## 5. Conclusions

In this study, the MODIS data, snow depth data, and four different CMIP6 snow cover data scenarios were used to calculate the SCF, SD, and SCDs of different SSP scenarios in the past and the future. The temporal and spatial characteristics of snow cover in the three stable snow cover areas were analyzed. The main conclusions of this study are as follows:

(1) In the past 20 years, the mean values of the SCF, SD, and SCDs were the highest in NX, at 37%, 3.43 cm, and 47.81 days, respectively. There was no statistically significant trend in the SCF, SD, and SCDs in the three stable snow cover areas. The seasonal variation of the TP is stable, and the snow reduction in NX and NC is mainly concentrated in spring and winter. SD and SCDs have opposite trends in areas with an elevation greater than 3000 m, and NC is less affected by elevation.

(2) The spatial distribution of the SCF, SD, and SCDs in the three stable snow cover areas is consistent and mainly distributed in the southeast and west of the TP, south and northeast of NX, and north of NC. The SCF in the three stable snow cover areas is mainly distributed between 20 and 40%, and the decreasing trend is the main trend in the areas with an SCF of >60%. The SD values of NX and NC accounted for large proportions of the areas above 4 cm, accounting for 27.6% and 27.1%, respectively, but the change trend of the two snow areas was the opposite. NX had the largest proportions of SCF and SCD reduction areas, with 24.6% and 57.6% of the area showing decreasing trends, respectively. The area of SD reduction in NC accounted for the largest proportion, with 58% of the area showing a decreasing trend. NX had the largest proportion of stable snow accumulation, and it showed an increasing trend.

(3) The future interannual changes in the three stable snow cover areas will continue to decline with the increase in development imbalance and showed a trend of first decreasing and then increasing under the SSPs126 scenario. Under the SSPs126 and SSPs245 scenarios, the response changes in the snow cover in the TP are the most significant, with the SCF decreasing by 15.3% and the SD by 33.3% by 2050. Under the SSPs370 and SSPs585 scenarios, the NX snow cover changed most significantly, with a 46.6% reduction in SD by 2100. Compared to the first half of the 21st century, the SCF, SD, and SCDs decreased significantly in the second half of the 21st century. Especially in the southeastern and western regions of the TP, the variation range of snow cover in the high development imbalance scenario was significantly larger than that in the low development imbalance scenario. Under the SSPs585 scenario, the SD decreased by 19.7 cm in the Karakoram Mountains, and the SCDs decreased by 2.5 day in the Nyenchentanglha Mountains.

(4) Future snow cover reductions are most pronounced in the southeast of TP, the northwest of NX, and the north of NC. As the development imbalance increased, the SCF and SD decreased the most in the TP, and the SCDs decreased the most in NX. Especially under the SSPs585 scenario, the SCF and SCD change rates of the TP reached $-1.15\%/10a$ and $-0.69$ cm/10a, respectively, and the SCD change rate of NX reached $-0.86$ day/10a.

**Author Contributions:** Conceptualization, Y.Z., P.S., Z.M., Y.L. and Q.Z.; methodology, Y.Z., P.S., Z.M., Y.L. and Q.Z.; validation, Y.Z., P.S., Z.M., Y.L. and Q.Z.; formal analysis, Y.Z., P.S., Z.M., Y.L. and Q.Z.; writing—original draft preparation, Y.Z., P.S., Z.M., Y.L. and Q.Z.; writing—review and editing, Y.Z., P.S., Z.M., Y.L. and Q.Z.; visualization, Y.Z., P.S., Z.M., Y.L. and Q.Z.; supervision, Y.Z., P.S., Z.M., Y.L. and Q.Z.; funding acquisition, P.S. and Q.Z.; All authors have read and agreed to the published version of the manuscript.

**Funding:** This research was funded by the Nature Science Foundation for Excellent Young Scholars of Anhui, grant number: 2108085Y13; the Key Research and Development Program Project of Anhui Province, China, grant number: 2022m07020011; The University Synergy Innovation Program of Anhui Province, grant number: GXXT-2021-048; Key projects of the support plan for outstanding young talents in Colleges and Universities, grant number: gxyqZD2021094; Major science and technology projects in Anhui Province, grant number: 202003a06020002.

**Data Availability Statement:** Not applicable.

**Acknowledgments:** We will thank for MODIS Science team for the MODIS data, and World Clmiate Research Programme for the CMIP6 data. We also thank MDPI for language assistance with the manuscript.

**Conflicts of Interest:** The authors declare no conflict of interest.

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
