# Peer review of "Snow Cover in the Three Stable Snow Cover Areas of China and Spatio-Temporal Patterns of the Future"

_remotesensing, doi:10.3390/rs14133098_

Round 1

Reviewer 1 Report

Lines 10 – 29 Acronyms should be limited in abstract

Keywords the first keyword is a phrase. Please look to appropriate glossaries! Check “snow cover days”: snow cover duration?

Line 46 SCF, SD and SCD must be declared here! SCD should be Snow Cover Duration?   8-10   51

Line 52 TP should be declared here, not in the abstract

Line 58-65 NC and NX should be declared here, not in the abstract

Figure 1 What is the box in lower right corner?

Section 2.2 is wrongly organized: 2.2.1 refers to satellite data and SD data, but snow cover data are useful for SCD and SCF estimations, not for SD….MODIS product should stays in 2.3 or be separated from SD

Section 2.3.3 Very difficult to follow, Eq 4-6 are disconnected from the text, even not cited. Does the Z value refer to Sen test?  I suppose not, but the sequence of equations let the reconstruction difficult.

Eq. 4 is based on what calculated in Eq 7, the order is wrong: Eq 7 is before

Eq. 5 what is sgn? Eq 6 should be before Eq 5

Eq. 7 is not formatted as an equation staying in one line

Eq 7 VAR(S) stays for variance?

Section 3 interpretation for figures 3 to 4 sounds speculative without statistical tests on stated changes. Figure 5 is not possible to be matched with comments since elevation and geographic areas are not reported in the figure.

Figure 6 is difficult to be interpreted, too many colours and symbols make confusion.

Section 4 presents a weak discussion that highlight the impact of the cloud cover on remotely sensed values without checking if detected trends are statistically significant. Model were considered without a comparison in the overlap period 2015-2020. It seems that models are disconnected in the manuscript structure  

Line 43 check typo: “[6]..”

Line 50 check typo “13].China”

Line 112 check typo “4.2×106km2” and “Figure 1.TP”

Line 176 The TIFF file format and the .tif file extension…..

Line 226 Check typo Figure 2. In bold

Lines 218-220 Should Z_* be un lowercase? Please read authors instructions!

Author Response

Thank you very much for your professional advice. For more detailed modification instructions, please refer to the attached Word.

Reviewer 2 Report

The Authors studied the areas covering almost 50% of China. The selected areas are important form the environmental, economical and agricultural point of view: northern Xinjiang (transregional economic belt); Qinghai-Tibet Plateau (water resources and supply) and northeast of China agriculture).

Three principial variables describing snow cover were analysed: snow cover fraction (SCF) known also as snow cover extent, snow depth (SD) and snow cover days (SCD). Tha study has based on good quality MODIS  snow cover products and both time series of changes and spatial distribution of snow in the period 2001 – 2002. The analysis does not raise any objection and the obtained results are important from the climatological point of view and interest.

The most interesting is a comparison of the snow variables in the light of the data taken from The Coupled Model Intercomparison Project (CMIP6) models. The Authors used 15 modes in CMIP6 snow cover data under 4 different fundamental scenarios of Socioeconomic Pathways forcing climate changes. The final results are important to apply the in farther studies and analyses of regional climate changes.

Author Response

Review report reviewer 2

The Authors studied the areas covering almost 50% of China. The selected areas are important form the environmental, economical and agricultural point of view: northern Xinjiang (transregional economic belt); Qinghai-Tibet Plateau (water resources and supply) and northeast of China agriculture).

Three principial variables describing snow cover were analysed: snow cover fraction (SCF) known also as snow cover extent, snow depth (SD) and snow cover days (SCD). Tha study has based on good quality MODIS snow cover products and both time series of changes and spatial distribution of snow in the period 2001 – 2002. The analysis does not raise any objection and the obtained results are important from the climatological point of view and interest.

The most interesting is a comparison of the snow variables in the light of the data taken from The Coupled Model Intercomparison Project (CMIP6) models. The Authors used 15 modes in CMIP6 snow cover data under 4 different fundamental scenarios of Socioeconomic Pathways forcing climate changes. The final results are important to apply the in farther studies and analyses of regional climate changes.

Reply:Thank you very much for your professional advice.

Reviewer 3 Report

This paper study the potential impacts of climate change on Snow Cover Area (SCA) in three areas of China. The subject is of potential interest for the Remote Sesing readership. The manuscript is well written and easy to read and follow. Nevertheless, in my opinion, the manuscript should address properly some issues to be published.  

  1. INTRODUCTION:

The introduction should be improved. The authors should make an effort to clarify the novelty of the paper, from a methodological point of view in a broader context and/or with respect to the case study. They should include a more clear definition of the knowledge gaps, indicating the main novelties of the proposed objectives and their relevance.

I also miss a more complete review of different approaches adopted to simulate climate change impacts on snow dynamic (ej. Collados-Lara et al., 2020; Wobus et al., 2017), and the pros and cons of them. For example, is the satellite information accurate enough to approximate Snow Depth? In which conditions? 

  1. DISCUSSION

I also miss a higher effort in their discussion, mentioning pros and cons, and similarities and differences with other approaches and results obtained when studying climate change impacts on snow dynamic.

I would suggest to include a subsection “assumption and limitations” to summarize limitations related to the adopted approach.

REFERENCES:

Collados-Lara AJ, Pardo-Igúzquiza E, Pulido-Velazquez D. 2019. A distributed cellular automata model to simulate potential future impacts of climate change on snow cover area. Advances in Water Resources, 124, 106-119. doi: 10.1016/j.advwatres.2018.12.010.

Wobus, C., Small, E.E., Hosterman, H., Mills, D., Stein, J., Rissing, M., Martinich, J., 2017. Projected climate change impacts on skiing and snowmobiling: a case study of the United States. Glob. Environ. Chang. https://doi.org/10.1016/j.gloenvcha.2017.04.006.

Author Response

Review report reviewer 3

This paper study the potential impacts of climate change on Snow Cover Area (SCA) in three areas of China. The subject is of potential interest for the Remote Sesing readership. The manuscript is well written and easy to read and follow. Nevertheless, in my opinion, the manuscript should address properly some issues to be published. 

  1. INTRODUCTION:

The introduction should be improved. The authors should make an effort to clarify the novelty of the paper, from a methodological point of view in a broader context and/or with respect to the case study. They should include a more clear definition of the knowledge gaps, indicating the main novelties of the proposed objectives and their relevance.

Reply:Thank you very much for your professional advice. It’s done. We hide a detailed description of the novel points of the paper in the last paragraph of the introduction.

Line 99-111

China has a vast territory, and the changes in snow cover between different regions are also different. In the past, most studies on China's snow cover focused on large-scale single indicators, small-scale multiple indicators, and CMIP5. There are few comprehensive studies on China's snow cover and future changes in CMIP6. Therefore, under the context of global warming, this study selects MODIS and CMIP6 data to analyze the temporal and spatial evolution of snow in the three stable snow cover areas in China. Assess the recent and future changes in snow cover in China. The objectives are to (1) use MODIS and passive microwave remote sensing data to analyze the main characteristics of the spatial pattern of SCF, SD and SCD in the three stable snow cover areas in China. (2) analyze the main characteristics of the spatial distribution and change trend of SCF, SD, and SCD in the future under different scenarios. (3) the historical and future changes of SCF, SD, and SCD in the three stable snow cover areas are compared. This study can be used as an important reference to provide scientific guidance for formulating the rational utilization of snow resources, future disaster risk assessment and regional planning in the three stable snow cover areas in China.

I also miss a more complete review of different approaches adopted to simulate climate change impacts on snow dynamic (ej. Collados-Lara et al., 2020; Wobus et al., 2017), and the pros and cons of them. For example, is the satellite information accurate enough to approximate Snow Depth? In which conditions?

Reply:Thank you very much for your professional advice. It’s done.

Line 77-87

Although remote sensing can easily and quickly obtain snow cover information, due to the limitation of time series length, rough spatial resolution and cloud layer interference, it is necessary to use models to invert snow cover. At present, there are many studies on snow model. Wobus et al. [35] used the Utah energy balance model to simulate the winter ski areas in the United States. The advantages of the UEB model are high computational efficiency, few input parameters and reliable results. The research shows that the UEB model has high accuracy in the simulation of snow depth in ski areas.  Collados et al. [36] used an improved cellular automata (CA) model to study the snow cover area of Sierra Nevada mountains. The CA model calculated the snow cover area through five parameters and two driving variables, and corrected each parameter to calculate the snow cover area. Adjusted to optimal, the results of the study suggest a significant reduction in snow cover in the region in the future.

Supplementary references:

35.Wobus, C.; Small, E.E.; Hosterman, H.; Mills, D.; Stein, J.; Rissing, M.; Jones, R.; Duckworth, M.; Hall, R.; Kolian, M.; Creason, J.; Martinich, J. Projected climate change impacts on skiing and snowmobiling: A case study of the United States. Glob. Environ. Chang. 2017, 45, 1-14, 10.1016/j.gloenvcha.2017.04.006.

36.Collados-Lara, A.J.; Pardo-Iguzquiza, E.; Pulido-Velazquez, D. A distributed cellular automata model to simulate potential future impacts of climate change on snow cover area. Adv. Water Resour. 2019, 124, 106-119, 10.1016/j.advwatres.2018.12.010.

  1. DISCUSSION

I also miss a higher effort in their discussion, mentioning pros and cons, and similarities and differences with other approaches and results obtained when studying climate change impacts on snow dynamic.

I would suggest to include a subsection “assumption and limitations” to summarize limitations related to the adopted approach.

Reply:Thank you very much for your professional advice. It’s done.

Line 524-529

General circulation model (GCM) are reliable at large scales and uncertain at regional scales. Sources of uncertainty include future emission scenarios, small regional climate change and model uncertainty [84]. The bias of the GCM itself also affects the reliability of the regional climate model (RCM) [85]. The bias correction based on the CMIP6 multimodal mean does not get good results in terms of snow depth. Related studies also show that CMIP6 overestimates SWE in the northern hemisphere [86].

Supplementary references:

84.Piao, J.; Chen, W.; Wang, L.; Chen, S. Future projections of precipitation, surface temperatures and drought events over the monsoon transitional zone in China from bias-corrected CMIP6 models. Int. J. Climatol. 2021, 1-17, 10.1002/joc.7297.

85.Xu, Z.; Han, Y.; Tam, C.; Yang, Z.; Fu, C. Bias-corrected CMIP6 global dataset for dynamical downscaling of the historical and future climate (1979–2100). Sci. Data. 2021, 8, 293, 10.1038/s41597-021-01079-3.

86.Kouki, K.; Räisänen, P.; Luojus, K.; Luomaranta, A.; Riihelä, A. Evaluation of Northern Hemisphere snow water equivalent in CMIP6 models during 1982–2014. Cryosphere. 2022, 16, 1007-1030, 10.5194/tc-16-1007-2022.
